# Depletion of Activated Hepatic Stellate Cells and Capillarized Liver Sinusoidal Endothelial Cells Using a Rationally Designed Protein for Nonalcoholic Steatohepatitis and Alcoholic Hepatitis Treatment

**DOI:** 10.3390/ijms25137447

**Published:** 2024-07-06

**Authors:** Falguni Mishra, Yi Yuan, Jenny J. Yang, Bin Li, Payton Chan, Zhiren Liu

**Affiliations:** 1Department of Biology, Georgia State University, Atlanta, GA 30303, USA; fmishra1@student.gsu.edu (F.M.); yyuan13@gsu.edu (Y.Y.); bli17@student.gsu.edu (B.L.); pchan2@student.gsu.edu (P.C.); 2Department of Chemistry, Georgia State University, Atlanta, GA 30303, USA; jenny@gsu.edu

**Keywords:** chronic liver disease, liver fibrosis, integrin α_v_β_3_, hepatic stellate cells, liver sinusoids, sinusoidal endothelial, collagen, myofibroblast

## Abstract

Nonalcoholic steatohepatitis (NASH) and alcoholic hepatitis (AH) affect a large part of the general population worldwide. Dysregulation of lipid metabolism and alcohol toxicity drive disease progression by the activation of hepatic stellate cells and the capillarization of liver sinusoidal endothelial cells. Collagen deposition, along with sinusoidal remodeling, alters sinusoid structure, resulting in hepatic inflammation, portal hypertension, liver failure, and other complications. Efforts were made to develop treatments for NASH and AH. However, the success of such treatments is limited and unpredictable. We report a strategy for NASH and AH treatment involving the induction of integrin α_v_β_3_-mediated cell apoptosis using a rationally designed protein (ProAgio). Integrin α_v_β_3_ is highly expressed in activated hepatic stellate cells (αHSCs), the angiogenic endothelium, and capillarized liver sinusoidal endothelial cells (caLSECs). ProAgio induces the apoptosis of these disease-driving cells, therefore decreasing collagen fibril, reversing sinusoid remodeling, and reducing immune cell infiltration. The reversal of sinusoid remodeling reduces the expression of leukocyte adhesion molecules on LSECs, thus decreasing leukocyte infiltration/activation in the diseased liver. Our studies present a novel and effective approach for NASH and AH treatment.

## 1. Introduction

Liver fibrosis/cirrhosis is caused by persistent inflammation responses to liver insults, which leads to the constant activation of HSCs and the apoptosis resistance of αHSCs [1,2,3,4,5,6]. HSC activation facilitates the trans-differentiation of HSCs into myofibroblast phenotypes [3,7]. αHSCs release high amounts of extracellular matrix proteins, mostly collagen, and MMP inhibitors, which leads to the accumulation of collagen bundles in the liver [8,9]. Excessive deposition of collagen bundles leads to fibrotic scarring, which disrupts the liver cytoarchitecture, resulting in liver function failure [10].

The vasculature in liver sinusoids differs from that in other organs. Sinusoids are made of liver sinusoidal endothelial cells (LSECs), which possess the unique feature of fenestration. In addition, liver sinusoids lack an organized basement membrane [11,12,13,14]. However, in fibrotic liver, responses to liver insults promote LSEC dedifferentiation or capillarization. LSEC capillarization, as well as LSEC proliferation and migration, leads to sinusoid remodeling and intrahepatic angiogenesis [15,16], resulting in vessel structure dysregulation in sinusoids [17,18]. The dysregulated vessel structure often leads to resistance to blood flow in the portal vein, especially in the sinusoidal space. The consequence is liver failure and other related complications, such as portal hypertension. Intact liver sinusoids are the protective gate that prevents passenger leukocytes from entering into the space of Disse due to a lack of expression of leukocyte adhesion molecules such as VAP-1, E-selectin, ICAM-1, LYVE-1, and PECAM-1 on healthy LSECs [19,20,21,22,23]. LSEC capillarization converts the immune-protective sinusoids into inflammation-stimulative sinusoids due to (1) the high levels of the leukocyte adhesion molecules expressed on caLSECs, allowing for leukocyte trans-endothelial migration [21,24,25], and (2) caLSECs’ secretion of pro-inflammatory cytokines and chemokines [21,25] that recruit and activate infiltrated leukocytes. Quiescent HSCs have minimal interaction with immune cells; therefore, they have a very limited role in modulating inflammation in the liver. Due to the liver damage response, αHSCs directly or indirectly contact various immune cells that enter into the space of Disse. αHSCs recruit and activate immune cells by releasing cytokines/chemokines [26,27,28], promoting the activation and survival of infiltrating leukocytes in the fibrotic liver [29]. The consequence is the exacerbation of induced inflammation in the liver [30].

Interestingly, the activation of HSCs, sinusoidal remodeling, and intrahepatic angiogenesis are tightly coupled. αHSCs strongly stimulate intrahepatic angiogenesis and sinusoidal remodeling. αHSCs secrete a number of molecular factors that promote LSEC growth migration and dedifferentiation [31,32], while caLSECs maintain HSC activation [33]. In addition, the ECM that is released by αHSCs also plays a role in promoting intrahepatic angiogenesis and sinusoidal remodeling [34,35]. On the other hand, caLSECs and intrahepatic angiogenesis, in turn, promote further HSC activation [31]. Thus, the activation of HSCs, sinusoidal remodeling, and intrahepatic angiogenesis form a vicious circle in facilitating liver fibrosis progression.

The progression of NASH and AH features the simultaneous accumulation of collagen fibrils and strong inflammation in the diseased liver: (1) αHSCs and caLSEC promote fibrosis progression; (2) more importantly, αHSCs and caLSECs in the fibrotic liver aggravate inflammation induced by liver damage by lipotoxicity and alcohol toxicity insults due to the important functions of αHSCs and caLSECs in the regulation of immune cell recruitment and activation in NASH and AH liver; (3) the progression of NASH and AH is characterized by hepatocyte loss due to disease-associated hepatocyte apoptosis. Quiescent HSCs and fenestrated sinusoids are the critical driving factors for liver regeneration [36,37,38,39]. It is clear that the continuous activation of HSCs, LSEC capillarization, and intrahepatic angiogenesis largely contribute to the failure of treatments for advanced NASH and AH. Given the functional importance of αHSCs and caLSECs in NASH and AH progression, the removal of αHSCs, reversal of LSEC capillarization, and elimination of intrahepatic angiogenesis could provide very important benefits for the disease’s treatment. Herein, we present a study on a rationally designed integrin α_v_β_3_-targeting protein (ProAgio) that depletes αHSCs and reverses sinusoidal remodeling in murine models of NASH and AH liver. The depletion of αHSCs by ProAgio leads to a reduction in the expression of collagen fibrils in NASH and AH liver. As a consequence of the removal of caLSECs and, thus, the reversal of LSEC capillarization, ProAgio decreases the infiltration of immune cells and, thus, inflammation in NASH and AH liver. ProAgio is a potentially effective agent for advanced NASH and AH treatment.

## 2. Results

### 2.1. ProAgio Decreases αHSCs and Reduces Fibrotic Collagen and Hepatic Inflammation in NASH Mouse Models

We previously reported the development of a protein (ProAgio) that targets integrin α_v_β_3_ at a non-ligand binding site. ProAgio induces the apoptosis of integrin α_v_β_3_, expressing cells by recruiting/activating caspase 8 to the cytoplasmic domain of the targeted integrin [40]. ProAgio reversed liver fibrosis/cirrhosis in TAA/alcohol-induced and CCl_4_-induced liver fibrosis murine models. Our study revealed that ProAgio simultaneously depleted αHSCs, resulting in a reduction in the expression of collagen fibrils, and reversed sinusoid remodeling, resulting in relieving blood flow resistance in the fibrotic liver of the mice in the murine models [41]. NASH represents a great health care challenge, as no effective strategies or agents are available for the treatment of NASH. Evidence suggests that both αHSCs and caLSECs play critical role in NASH development and progression [42,43]. We sought to identify whether ProAgio would be effective in NASH treatment due to its unique drug activity. We employed the common NASH diet-induced NASH mouse model [29,44]. Nine-week-old C57BL/6 mice were fed a high-fat (HD) NASH diet and d-glucose in drinking water. After 29 weeks of NASH diet feeding, the animals were treated via ProAgio 10 mg/kg or a vehicle daily, with i.p. dosing, for either 4 days (early treatment, ET) or 2 weeks (late treatment, LT). The animals were euthanized 3 days after the last dose of the ET or LT treatment (Figure 1A). The mice were continuously on a NASH diet during the ProAgio or vehicle treatment. Sirius Red staining of liver sections demonstrated less and thinner collagen accumulation in the livers of the ProAgio-treated animals compared to the vehicle-treated animals (Figure 1B,C). IHC staining of αSMA in liver sections showed reduced αSMA-positive αHSCs in NASH liver (Figure 1D,E). The primary driver of NAFLD/NASH is over-nutrition. Therefore NAFLD/NASH is associated with an alteration in blood glucose levels [45,46]. We sought to identify whether ProAgio treatment would improve blood glucose in the NASH mice. The C57BL/6 mice were fed a high-fat NASH diet for 22 weeks. The mice were treated with ProAgio 10 mg/kg i.p. daily dose for 10 days or a vehicle treatment. Two weeks after the treatment, a glucose tolerance test was performed on the treated animals. Apparently, ProAgio lowered the fasting blood glucose and improved the glucose tolerance of the NASH mice (Figure 1F). H&E staining of the liver sections indicated the clearance of fat accumulation in the liver (Figure 1G). Our results suggest that ProAgio is effective in the HD-NASH diet-induced murine NASH model.

One typical feature of NASH progression is hepatocyte ballooning due to inflammation-induced hepatocyte injury [47,48]. Close examination of the H&E staining results revealed that ProAgio treatment clearly reduced hepatocyte ballooning (the vehicle-treated mice score went from scoring 2 to scoring 0, similar to that of normal mice) (Figure 1G), suggesting that the treatment might decrease inflammation in NASH liver. To verify the effects of ProAgio on NASH liver inflammation, IHC F4/80 staining in the liver sections showed reduced macrophages in NASH liver (Figure 2A,B). Furthermore, FACS analysis of innate immune cells also verified the effects of ProAgio on the infiltration/activation of macrophages and polymorphonuclear neutrophils (PMNs) in NASH liver (Figure 2C,D). It has recently been demonstrated that CD44 is a key player of hepatic inflammation during NASH and liver fibrosis progression [49,50]. We therefore explored the effects of ProAgio on CD44 in NASH liver. IHC staining of CD44 with the liver sections showed that ProAgio decreased the CD44 levels by 1.5-fold compared to the vehicle-treated group (Figure 2E,F). Our experiments clearly suggest that ProAgio strongly reduced hepatic inflammation associated with NASH progression, providing an important treatment advantage.

### 2.2. ProAgio Is Effective with Alcoholic Hepatitis Model

A common shared feature of both NASH and alcoholic hepatitis (AH) is the accumulation of collagen fibers and alleviated inflammation in the diseased liver during disease progression. We thus questioned whether ProAgio would exert similar effects in the context of AH. To this end, a high-fat diet plus binge alcohol-induced AH model [44,51] was employed to test the effectiveness of ProAgio. Nine-week-old C57BL/6 mice were fed a high-fat diet for 90 days. Binge alcohol (5 g/kg) was given to the animals (via oral gavage) twice weekly for four weeks. ProAgio treatment started after 3 weeks of ethanol gavage for 12 daily doses (Figure 3A). Liver sections were prepared from 6 mice/group 1 day after the last ProAgio treatment. Sirius Red staining of liver sections demonstrated less collagen accumulation in the livers of the ProAgio-treated mice (Figure 3B,C). IHC staining of α-SMA with the liver sections showed a decrease in αHSCs (Figure 3D,E). It is well established that CD44 is a key player of hepatic inflammation during AH and liver fibrosis progression. We therefore also analyzed the effects of ProAgio on CD44 in AH liver. IHC staining of CD44 with the liver sections showed that ProAgio decreased the CD44 levels by 2.5-fold compared to the vehicle-treated group (Figure 3F,G). To test whether ProAgio also decreased leukocyte infiltration/activation in AH mice as we observed in NASH liver, we analyzed immune cells in AH liver. FACS analysis of Ly6G in liver extracts suggested that ProAgio reduced neutrophils in AH liver by >3-fold (Figure 4A). F4/80 staining showed that macrophages decreased by >2.5-fold upon ProAgio treatment (Figure 4B,C). Our results provided proof that ProAgio is effective in the treatment of advanced AH in a murine model.

### 2.3. ProAgio Reverses Sinusoidal Remodeling and Reduces Leukocyte Adhesion Molecules on LSECs

The question of how ProAgio decreases leukocyte infiltration in both NASH and AH liver is an open question. Intact liver sinusoids are the protective gate that prevents passenger leukocytes from entering into the space of Disse due to a lack of expression of leukocyte adhesion molecules on healthy LSECs [19,20,21,22,23]. LSEC capillarization converts the immune-protective sinusoids into inflammation-stimulative sinusoids due to the high levels of the leukocyte adhesion molecules expressed on caLSECs, allowing for leukocyte trans-endothelial migration [21,24,25]. We previously reported that ProAgio reversed sinusoid remodeling, resulting in the relief of blood flow resistance in fibrotic liver of mice in TAA/alcohol- and CCl_4_-induced liver fibrosis murine models [41]. Thus, we sought to examine whether ProAgio also reversed sinusoid remodeling and LSEC capillarization in the NASH and AH livers. We analyzed the effects of ProAgio on sinusoids in NASH and AH liver. Similar to our previous study, IHC staining of SE-1 revealed that ProAgio treatment increased SE-1 staining (Figure 4D,E), thus decreasing caLSEC expression. These analyses suggested that ProAgio reversed sinusoidal remodeling and LSEC capillarization. Next, we analyzed leukocyte adhesion molecule expression in the NASH and AH liver upon ProAgio treatment. IHC staining of VAP-1 and LYVE-1 [52,53] in the sections from both the NASH and AH livers demonstrated that the levels of LYVE-1 and VAP-1 were increased due to the induction of NASH and AH. ProAgio treatment dramatically decreased LYVE-1 and VAP-1 in the NASH and AH liver to levels similar to those in normal healthy liver (Figure 5A–D). Our results suggest that ProAgio reversed LSEC capillarization and sinusoidal remodeling in NASH and AH livers. The reversal of sinusoidal remodeling decreased the expression of leukocyte adhesion molecules on the LSECs, thus preventing leukocyte infiltration and decreasing hepatic inflammation.

## 3. Discussion

The expression of integrin α_v_β_3_ is elevated upon HSC activation [54,55,56]. Although, the function(s) of this up-regulation in HSC activation is not fully understood, it was speculated that targeting this integrin using the integrin ligand mimics cilengitide might be a good approach for targeting αHSCs for liver fibrosis. Unfortunately, although cilengitide exhibited effects in inhibiting the proliferation of αHSCs in vitro, experiments with a mouse model indicated that cilengitide actually exacerbated the disease. Analyses showed that cilengitide did not induce apoptosis of αHSCs in the fibrotic liver [54,55]. In addition to their strong pro-fibrogenic role, αHSCs also play a direct role in vascular structure changes in fibrotic liver. HSCs act as vascular contractile machinery in response to vasoconstrictors and vasodilators. However, in fibrotic liver, the function of the vascular contractor of HSCs is dysregulated due to the activation of HSCs [12,33,57]. In addition to secreting collagen fibrils, αHSCs also play a direct role in the activation of hepatic inflammation by the release of inflammation-promoting cytokines and chemokines during CLD progression [28,58]. Accompanying CLD progression, the dedifferentiation or capillarization of LSECs leads to liver blood vessel alterations. The capillarization of LSECs is the key player in converting sinusoids from immune-protective gates to inflammation stimulators [59]. Furthermore, intact healthy LSECs are a key driver in hepatic regeneration [21,39]. Thus, the targeted depletion of αHSCs and caLSECs would be critically important for removing collagen fibrils, reducing hepatic inflammation, and facilitating liver regeneration in CLD, including NASH and AH, treatment. However, in CLD liver, the activation of HSCs and capillarization of LSECs is tightly coupled. This tightly coupled regulation makes one event affect the other one. Therefore, by targeting one event individually, it may be less effective. ProAgio targets a common molecular signature integrin, α_v_β_3_, of αHSCs and caLSECs without producing effects on quiescent HSCs and LSECs. By this dual targeting mechanism, ProAgio simultaneously and specifically eliminates disease-causing cell types, αHSCs and, caLSECs. ProAgio depletes αHSCs, therefore reducing collagen accumulation in NASH and AH liver. ProAgio removes caLSEC, therefore reversing sinusoid remodeling, and eliminates hepatic angiogenesis. The removal of caLSECs decreased the expression of immune cell attachment molecules on sinusoids. Through this action, along with the depletion of αHSCs, ProAgio reduced immune cell infiltration in NASH and AH liver. Furthermore, the reversal of sinusoid remodeling greatly promotes hepatic regeneration, which is an important aspect in NASH and AH treatment (Figure 5E). The dual targeting mechanism certainly brings a very important advantage of breaking down the vicious cycle of angiogenesis–fibrogenesis in CLD progression. Extensive toxicity tests of ProAgio with healthy animals, including mice, rats, and monkeys, and cancer patients demonstrated that ProAgio is well tolerated at a high dose. It will be important to test whether ProAgio exerts strong toxic effects on NASH and AH animal models before moving into clinical studies of ProAgio with NASH and AH patients.

## 4. Methods

### 4.1. NASH and AH Induction and Treatments

All animal experiments were carried out in accordance with the guidelines of NIH and approved by the IACUC of Georgia State University. Eight-week-old C57BL/6 mice, equal m/f, were fed with food containing a high fat content and long-chain trans-fats. High sucrose corn syrup and cholesterol were added to the diet [(Teklad diets, Envigo, Madison, WI, USA, TD. 120528), supplemented with (1.25%) cholesterol and (41%) sucrose. The mice were also provided 23.1 g/L d-fructose and 18.9 g/L d-glucose in drinking water]. The animals were fed for 25–29 weeks to induce NASH and fibrosis (NASH fibrosis model). Eight-week-old C57BL/6 mice were fed with food (Teklad diets, Envigo, Madison, WI, USA, TD.06414)containing a high fat content and long-chain trans-fats (60% fat). After 90 days of high-fat diet feeding, ethanol (5 g/kg body weight) was administered to the animals twice per week for 4 weeks via oral gavage. During ethanol gavage, the animals were kept on high-fat diet feeding. The NASH and AH mice were subjected to ProAgio treatment via daily i.p. injection. The animals continued to consume a high-fat diet, and NASH and AH were induced during the treatments.

### 4.2. Tissue Section Staining

**Sirius Red:** Sirius Red staining was carried out using NovaUltraTM Sirius Red Stain Kit from IHC WORLD by following the instructions of vendor.

**Immunohistochemistry:** The IHC staining procedures were similar to those of previous reports [40,60]. Images were captured at a 20× lens aperture, and the scale bars indicate 50 μM in length. Three–six view fields per section were evaluated per sample. The positively stained area was analyzed densitometrically as the area stained per total tissue area and normalized to total tissue area using Fiji software version 2.14.0/1.54f. 

**FACS, immunoblots, hydroxyproline, and proliferation assays:** Procedures similar to those used in our previous report [41] were followed for these assays.

### 4.3. Statistical Calculations

Data were statistically analyzed by comparing two appropriate groups. *p* values were calculated using an unpaired two-tailed Student *t*-test. In all figures and tables, NS *p* > 0.05, and statistical insignificance was set as follows: * *p* < 0.05, ** *p* < 0.01, and *** *p* < 0.001.

## Figures and Tables

**Figure 1 ijms-25-07447-f001:**
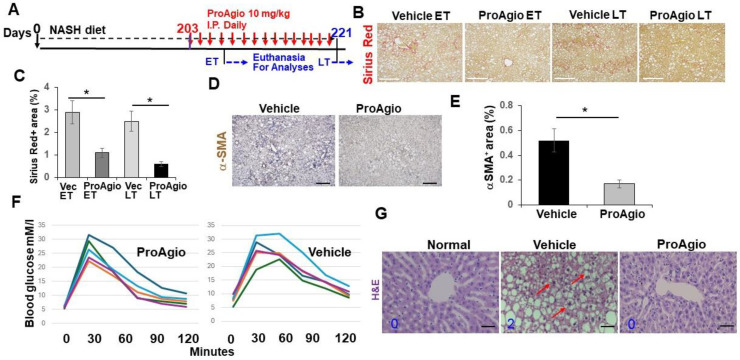
ProAgio decreases hepatic fibrosis in NASH mice. (**A**) The scheme illustrates the ProAgio or vehicle treatment regimen of the NASH mice. (**B**,**C**) Representative images (**B**) and quantifications (**C**) of Sirius Red staining of liver sections from the treated mice. ET, early treatment; LT late treatment. (**D**,**E**) Representative images (**D**) and quantifications (**E**) of IHC staining of αSMA in liver sections from the treated mice. Quantitation of collagen by Sirius Red and αSMA levels by IHC of αSMA staining using Fiji software version 2.14.0/1.54f. Four randomly selected tissue sections per animal, three randomly selected view fields in each section, and six randomly selected animals (n = 6) were quantified. The quantity of collagen and αSMA levels are presented as % of staining positive area. (**F**) Blood glucose levels (mM/L) of five mice (n = 5, showing in different color) treated with the indicated agents were measured at indicated time points before (time 0) and after i.v. injection of 2 g/kg glucose. The mice were fasted overnight before the glucose injection and measurements. (**G**) Representative images of H&E-stained liver sections from NASH mice treated with the indicated agents. Hepatic ballooning was scored by a hepatic pathologist based on H&E staining (see the arrows in (**G**) for examples; 5 randomly selected sections per animal were scored n = 5): normal and ProAgio-treated animals score 0; vehicle-treated animals score 2 (number in each panel). The error bars in (**C**,**E**) are the standard deviations of five independent mice. Statistical analysis of data was performed by a one-way Student’s *t*-test. (* *p* < 0.05).

**Figure 2 ijms-25-07447-f002:**
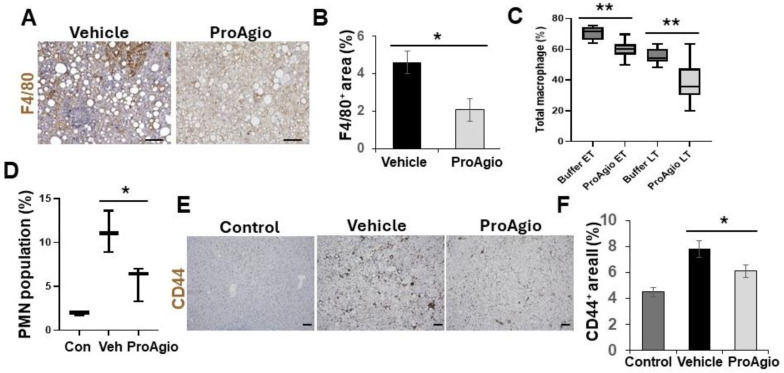
ProAgio reduces inflammation in NASH mouse liver. (**A**,**B**) Representative images (**A**) and quantifications (**B**) of F4/80 staining of liver sections from mice treated with indicated agents. The quantity of total macrophage levels is presented as % of staining positive area. (**C**,**D**) Total population (%) of macrophages (**C**) and polymorphonuclear neutrophils (PMNs) (**D**) in liver tissues from NASH mice treated with indicated agents were analyzed by FACS. For macrophages in (**C**), CD45^+^/CD11b^+^/F4/80^+^/Ly-6C^−^ cells were used. For PMN in (D), CD45^+^/CD11b^+^/Ly-6G^+^ were used. (**E**,**F**) Representative images (**E**) and quantifications (**F**) of IHC staining of CD44 in liver sections from NASH mice treated with indicated agents. The quantity of CD44 levels is presented as % of staining positive area. Con in (**D**) and Controls in (**E**,**F**) means the mice were normal healthy mice without any disease induction and subsequent treatment. The error bars in (**B**–**D**,**F**) are standard deviations of measurements of 4 mice for (**B**,**C**) and 6 mice for (**D**,**F**). Statistical analysis of data was performed via a Student’s *t*-test for a two-group comparison by a one-way ANOVA with Tukey’s multiple comparison test. (* *p* < 0.05, ** *p* < 0.01).

**Figure 3 ijms-25-07447-f003:**
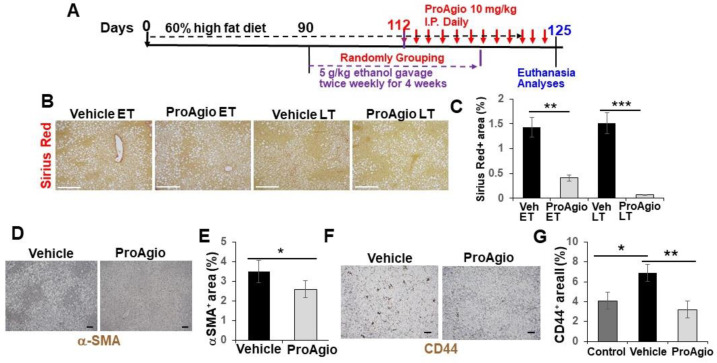
ProAgio decreases hepatic fibrosis in AH mice. (**A**) The scheme illustrates the ProAgio or vehicle treatment regimen for the AH mice. (**B**,**C**) Representative images (**B**) and quantifications (**C**) of Sirius Red staining of liver sections from mice treated with indicated agents. (**D**,**E**) Representative images (**D**) and quantifications (**E**) of IHC staining of αSMA in liver sections from the treated mice. (**F**,**G**) Representative images (**F**) and quantifications (**G**) of IHC staining of CD44 in liver sections from mice treated with indicated agents. The quantities in (**C**,**E**,**G**) are presented as % of staining positive area. Three randomly selected view fields per section, four randomly selected sections per animal, and six randomly selected animals (n = 6) were quantified. Veh in (**G**) is the vehicle-treated group. Control in (**G**) means the mice were normal healthy mice without disease induction and subsequent treatments. The error bars in (**C**,**E**,**G**) are standard deviations of 6 independent mice. Statistical analysis of data was performed by a Student’s *t*-test for a two-group comparison or a one-way ANOVA with Tukey’s multiple comparison test. (* *p* < 0.05, ** *p* < 0.01, and *** *p* < 0.001).

**Figure 4 ijms-25-07447-f004:**
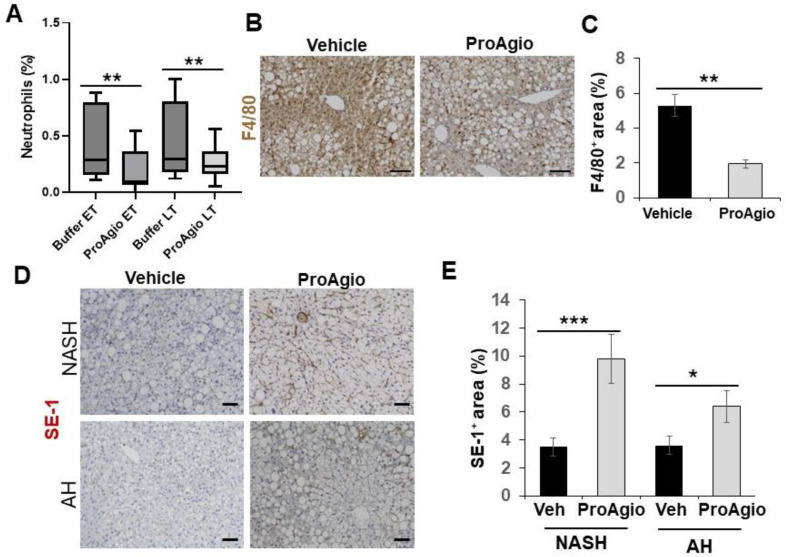
ProAgio reduces inflammation in AH mouse liver. (**A**) The total population of neutrophils (Neutrophil%) in the liver tissues from the AH mice that were treated with indicated agents was analyzed by FACS (CD11b^+^ Ly-6G^+^). (**B**,**C**) Representative images (**B**) and quantifications (**C**) of F4/80 staining of liver sections from mice treated with indicated agents. The quantity of total macrophage levels is presented as % of staining positive area. (**D**,**E**) Representative images (**D**) and quantifications (**E**) of IHC staining of SE-1 in liver sections from the NASH and AH mice that were treated with indicated agents. The quantity of differentiated healthy LSEC levels in (**E**) is presented as % of SE-1-positive staining area. The error bars in (**A**,**C**,**E**) are standard deviations of measurements of 6 mice. Statistical analysis of data was performed by a one-way ANOVA with Tukey’s multiple comparison test or an unpaired Student’s *t*-test for two-group comparisons. (* *p* < 0.05, ** *p* < 0.01, and *** *p* < 0.001).

**Figure 5 ijms-25-07447-f005:**
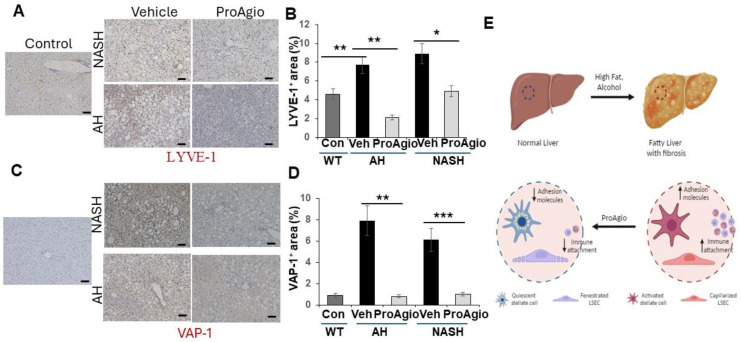
ProAgio decreases immune cell attachment molecules in NASH and AH mice (**A**–**D**) Representative images (**A**,**C**) and quantifications (**B**,**D**) of IHC staining of LYVE-1 (**A**,**B**) or VAP-1 (**C**,**D**) of liver sections from NASH or AH mice treated with indicated agents. The total LYVE-1 or VAP-1 levels are presented as % of staining positive area. Con and control mean the sections from normal healthy mice without any disease induction or subsequent treatment. (**E**) Scheme illustrating the drug actions of ProAgio in steatohepatitis. The error bars in (**B**,**D**) are standard deviations of measurements of 6 mice. Statistical analysis of data was performed by a one-way ANOVA with Tukey’s multiple comparison test. (* *p* < 0.05, ** *p* < 0.01, and *** *p* < 0.001).

## Data Availability

The raw data supporting the conclusions of this article will be made available by the authors on request.

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
