# Peer review of "Depletion of Activated Hepatic Stellate Cells and Capillarized Liver Sinusoidal Endothelial Cells Using a Rationally Designed Protein for Nonalcoholic Steatohepatitis and Alcoholic Hepatitis Treatment"

_ijms, 2024, doi:10.3390/ijms25137447_

Round 1

Reviewer 1 Report

Comments and Suggestions for Authors

Authors showed possible effectiveness of ProAgio for the treatment of NASH or AH. No definitive treatment for NASH or AH has been established. Therefore, present study is valuable and interesting. Continuous administration of ProAgio with ip for about 2 weeks decreased aHSC or macrophage and improved liver fibrosis. These effects were clearly shown. But several issues require to be addressed for clinical application.

1. Adverse effects should be considered during this therapy. Compromised condition might be considered. Body weight, biochemical and hematological disorder should be checked.

2. Treatment duration or dose should be optimized. Authors should describe or discuss them.

3. Unsolved issues should be discussed for clinical application.  

Reviewer 2 Report

Comments and Suggestions for Authors

This study investigated the treatment effect of ProAgio for NASH and AH. The authors showed that ProAgio could reduce collagen fibrils and inflammation in NASH and AH.  They provided a novel strategy for NASH and AH treatment. While this is an interesting study, it would be perfect if the following concerns could be addressed.

1.    Section 4.1 could report the sample size to indicate how many mice you used for each group. Section 4.2 mentioned that you used IF, however, I couldn’t find IF results in Figures.

2.    In section 4.3, it was mentioned that the t-test was used to compare the difference between two groups. It seems like you have also compared multiple groups, such as in Figure 2F, 3G, 5D, and 5B. Have you adjusted the p-value when conducting hypothesis test for multiple groups?

3.    Can you please put the explanation in figure legends for the quantification of IHC?   For example, on the Y-axis “% of staining positive area”, does the “area” refer to the cells in a randomly selected view that are all positive with strong staining intensity, or in a percentage. Not sure if that is an IHC score. Could you provide how many samples were involved in calculating the %?

4.    Figure 5E shows the ProAgio in lung cancer based on the figure legend. Also, the resolution of some IHC images is low.

Round 2

Reviewer 1 Report

Comments and Suggestions for Authors

Authors properly responded to the reviewer's comments.

Reviewer 2 Report

Comments and Suggestions for Authors

Comments were addressed.